# The Response to Inoculation with PGPR Plus Orange Peel Amendment on Soybean Is Cultivar and Environment Dependent

**DOI:** 10.3390/plants11091138

**Published:** 2022-04-22

**Authors:** Maria Letícia Pacheco da Silva, Francesco S. Moen, Mark R. Liles, Yuchen Feng, Alvaro Sanz-Saez

**Affiliations:** 1Department of Crop, Soil and Environmental Sciences, Auburn University, Auburn, AL 36849, USA; mzp0109@auburn.edu (M.L.P.d.S.); fengyuc@auburn.edu (Y.F.); 2Department of Biological Sciences, Auburn University, Auburn, AL 36849, USA; fsm0005@auburn.edu (F.S.M.); lilesma@auburn.edu (M.R.L.)

**Keywords:** cultivar variation, orange peel, nitrogen fixation, photosynthesis, plant growth promotion rhizobacteria, root growth, yield

## Abstract

Plant growth-promoting rhizobacteria (PGPR) effects on plant yield are highly variable under field conditions due to competition with soil microbiota. Previous research determined that many *Bacillus velezensis* PGPR strains can use pectin as a sole carbon source and that seed inoculation with PGPR plus pectin-rich orange peel (OP) can enhance PGPR-mediated increases in plant growth. Because the previous studies used a single soybean cultivar, the objective of this research was to test the effect of PGPR plus OP inoculation on plant responses in a wide range of soybean cultivars. Preliminary screening with 20 soybean cultivars in the greenhouse showed that the PGPR plus OP produced a positive increase in all plant growth parameters when all cultivar data was averaged. However, when the inoculation response was examined cultivar by cultivar there was a range of cultivar response from a 60% increase in growth parameters to a 12% decrease, pointing to the presence of a cultivar-PGPR specificity. Further greenhouse and field experiments that studied cultivars with contrast responses to synbiotic inoculation revealed that the environment and/or the molecular interactions between the plant and microorganisms may play an important role in plant responsiveness.

## 1. Introduction

The world population is expected to grow to almost 9.73 billion by 2050 [1] at an annual rate of 77 million people per year [2], which will drive the need for increased food production. To meet the projected global food and fiber demands for a growing population by 2050, current crop production will need to double [3] requiring a ~2.4% growth rate per year [4]. However, the biggest challenge for modern agriculture is to increase plant productivity in an environmentally sustainable manner [5]. Soil infertility is a major crop yield constraint in developing nations [6]. Chemical fertilizers are the principal input used to increase soil fertility and crop yield. However, excessive use of chemical fertilizers leads to environmental pollution and soil structure degradation [4]. In this context, there is a need for technologies to sustainably improve nutrient absorption by plants and reduce the use of chemical fertilizers [7].

Soybean and other legumes are important components of crop rotations due to the ability to promote nitrogen fertility via biological nitrogen fixation (BNF) via diazotrophic symbionts such as *Bradyrhizobium japonicum* [8] that provide an abundant source of biologically available nitrogen for plants [9]. In addition, biofertilizers, substances that contain viable microorganisms capable of enhancing nutrient uptake and transportation by plants when applied to the seeds or soil [10], are an environmentally friendly and cost-effective source of plant nutrients that can promote sustainable crop production [6]. Plant Growth Promotion Rhizobacteria (PGPR) have been used as biofertilizers either by helping to provide nutrients to the plants or by influencing plant growth [11]. 

PGPRs are microorganisms that evolved plant associations [12] and consume root exudates released by the plant host and in return secrete metabolites (e.g., root hormones such as indole-3-acetic acid) that can promote plant growth [13]. For example, bacterial strains from the genera *Pseudomonas*, *Azospirillum*, *Azotobacter*, *Bacillus*, *Klebsiella*, *Enterobacter*, *Xanthomonas*, *Arthrobacter*, *Burkhoolderia*, *Paenibacillus* and *Serratia* have been shown to have PGPR activity [14,15]. The genus *Bacillus* in particular have been studied as PGPR due to their advantageous physiological traits, such as the ability to form hardy spores, which contributes to their survival in soils and a long shelf life that is conducive to field applications [16]. Many *Bacillus* spp. strains have been identified as PGPR with commercial potential use as biofertilizers and biocontrol agents [17]. Within this genus, strains of *B. velezensis* (Bv; previously known as *Bacillus amyloliquefaciens* subsp. *plantarum*) such as Bv BAC03 [18] and Bv FZB42 exert plant growth-promoting activities through efficient colonization of plant roots [19]. 

Most of the experiments with PGPRs have been performed under controlled environmental conditions, such as greenhouses. In addition, these studies are usually performed in small pots where the soil can be easily sterilized to ensure that the introduced PGPR gets established into the rhizosphere. However, when PGPRs are used in agricultural soils under field conditions their efficacy is more variable and even inexistent [20]. The lack of consistent success suggests that the introduced microbial population declines rapidly after inoculation [21,22]. The decline in viability, numbers and/or metabolic activity of the PGPR population is thought to be caused by the inability of the inoculated PGPR to compete with the native microbiota of that specific soil [23] and/or by the decrease in the PGPR population and activity due to environmental factors [24]. To observe significant plant growth promotion, it has been estimated that it requires a PGPR population of 10^3^–10^4^ cells per gram of root tissue [15]. PGPRs that are adapted to agricultural soils typically are observed to have a faster growth rate and an ability to metabolize various natural and xenobiotic compounds [25]. Furthermore, there are plant cultivar-specific differences that can select for or against rhizosphere microbial populations [5,26].

Previous research determined that PGPR strains of Bv commonly can use pectin as a sole carbon source, and that soybean seed co-inoculation with Bv and purified pectin or pectin-rich orange peel (a “synbiotic” that consists of a prebiotic and probiotic) can promote soybean growth and nodulation [27]. Pectin is a complex polysaccharide first isolated and described by Henri Braconnot in 1825 [28] which consists primarily of galacturonic acids and is present in the primary cell walls and middle lamella in plants [20,29]. While purified pectin has been demonstrated to increase the survival of some Bv strains and promote soybean growth, it is not a cost-effective source of pectin for use as a seed amendment. Therefore, other less expensive, pectin-rich amendments were investigated for compatibility with PGPRs to increase soybean growth. Pectin is present in most of the plant tissues, but apple pomace and orange peel are the two most important sources of industrial pectin [30]. Citrus crops are among the most widely cultivated fruits and more than 80% of citrus is manufactured to obtain juice, jams, jellies, etc. [31]. After the industrial extraction of orange juice, large amounts of orange peel remain as a by-product [32]. Orange peel (OP) is a good source of pectin as it contains up to 52.9% pectin content [33] and has been identified to be used by PGPR strains such as pectinolytic Bv AP193 [27]. Therefore, to increase the sustainability of agricultural systems by using PGPRs as biofertilizers, it is important to research new strategies to improve PGPR efficacy in agricultural soils in combination with a cost-effective pectin-rich amendment such as orange peel. The primary objective of this study was to test the responsiveness of different soybean cultivars to seed treatment consisting of PGPR and orange peel.

## 2. Material and Methods

### 2.1. Preliminary Experiment to Test Soybean Cultivar Variation to Orange Peel Amendments

In order to test whether all soybean cultivars respond the same to inoculation with Bv AP193 and orange peel (OP), a greenhouse experiment with 20 soybean cultivars was performed in December of 2019. 

#### 2.1.1. PGPR Strains, Soybean Cultivars and Growing Conditions 

As a substrate for the experiment, Sandy Loam field soil was collected from E.V. Smith Research Center in Shorter, AL, USA, specifically from fields with a history of soybean cultivation to ensure viable populations of *B. japonicum*. Standard Classic 400 pots (3.8 L) were prepared with fabric mesh in the bottom and filled with 4.7 kg of moist soil. Twenty commercially available cultivars were used for this experiment (Appendix A). Soil analysis performed at Auburn University Soil Testing Laboratory indicated that the soil had a pH of 6.35 and a composition of 12.3 kg ha^−1^ P, 112 kg ha^−1^ K, 113 kg ha^−1^ Mg, 425 kg ha^−1^ Ca. As recommended for soybean production, the equivalent of 89.6 kg ha^−1^ of P_2_O_5_ was added to each pot. For this calculation, the area of the top of the pot was calculated (0.19 m^2^) and the amount of fertilizer per pot was calculated. 

Seeds were surface sterilized in a 2% sodium hypochlorite solution and then washed several times in sterile water to remove chlorine residues as described in Sanz-Saez et al. [34]. Two treatments were evaluated in this experiment: Uninoculated and inoculated seeds with Bv AP193 plus OP, with four replications in total for each treatment. Bv AP193 spores were prepared following the methods of Hassan et al. [27] and added to each seed at a final concentration of 1 × 10^6^ spore colony forming units (CFUs) in 50 µL of sterile water. OP powder (Citrus Extracts LLC, Fort Pierce, FL USA) was used to prepare the OP suspension at a final concentration of 10 mg/200 µL per seed. 

Treatments were applied on seeds at sowing time. Five seeds were evenly placed 2.5 cm below the soil surface of each pot to ensure germination. Each seed, in the inoculated treatment group, was inoculated first with 200 μL of OP powder solution and then with 50 µL of Bv AP193. The seeds in the uninoculated treatment group received 250 µL of sterile water. Soil was moist at the time of planting and no water was added to either treatment group for at least 24–48 h to allow the seeds in the inoculated treatment group to remain in contact with the Bv and OP suspension. After emergence (approximately one week after sowing), only one seedling was kept per pot. 

Pots were aligned in rows of four with four pots per row (16 per table) and rearranged in a randomized complete block design within each repetition. The pots were rotated around the tables on the greenhouse each week, preventing any biases based on pot location and light intensity among pots. Artificial LED light (800 μmol mol^−1^ PAR) was used to maintain a photoperiod of 14 h of light and 8 h of darkness. Temperatures in the greenhouse ranged from 18 to 25 °C during the day and 10 to 20 °C at night. Each pot received 500 mL of water every 2 days. Each week plants were sprayed with Tundra EC (Winfield/AgriSolutions, Albertville, MN, USA), Talstar-Pro (FMC Inc., Philadelphia, PA, USA), and Kontos (OHP, Inc., Bluffton, SC, USA) to prevent and control insect infestation.

#### 2.1.2. Physiological and Growth Parameters Measurements

When the plants reached the R2 growth stage (flowering, ~30 days after planting, Fehr et al.,1971), SPAD values, a proxy for chlorophyll concentration, were measured using a SPAD-502 (Minolta, Tokyo, Japan). After that, plants were harvested, and total aboveground biomass (g plant^−1^) was calculated by separating leaves and stems and drying them at 60 °C for at least 72 h and then weighing them on a precision scale. Before drying, total leaf area (cm^2^ plant^−1^) was calculated by passing each trifoliated leaf through an LI-3000 Leaf Area Meter (LI-COR Biosciences, Lincoln, NE, USA). 

Roots were cleaned after harvesting using tap water and the nodules were separated. Fresh nodules were cleaned and placed over a clean white paper and were imaged with a digital camera. The pictures were analyzed for quantitative nodule characteristics using the ImageJ software, the same way as used in Riedell et al. [35], and nodule number and size (total cm^2^ plant^−1^ and individual nodule cm^2^ plant^−1^) were calculated. The imaged nodules were dried at 60 °C for at least 72 h to determine total nodule dry weight (g plant^−1^). Cleaned roots were scanned in a Winrhizo desk top scanner (Regent Instruments Inc., Sainte-Foye, Quebec, Canada) to calculate total root area (cm^2^ plant^−1^), root width (cm plant^−1^), root height (cm plant^−1^) and total root length (cm plant^−1^). After scanning the roots, they were dried at 60 °C for at least 72 h to calculate total root dry weight (g plant^−1^). 

#### 2.1.3. Statistical Analysis

A two-way ANOVA was performed for each parameter to test the effect of inoculation (non-inoculated control and Bv AP193 plus OP), cultivars (Table 1) and their interaction. A two-way ANOVA, with inoculation and cultivars as main factors and replication as random variable, was performed using PROC GLIMMIX in SAS (SAS 9.4, SAS Institute, Cary, NC, USA). When the main effect of inoculation and/or cultivar, or their interaction was significant, the least square means post hoc test was performed to compare means (LSMEANS, SAS 9.4, SAS Institute, Cary, NC, USA).

### 2.2. Green House Experiment to Test the Response of Inoculation with PGPR Plus Orange Peel Amendment on Contrasting Soybean Cultivars

#### 2.2.1. PGPR Strains, Soybean Cultivars and Growing Conditions

A greenhouse experiment was established at Auburn University from March to May 2020 with three commercial soybean cultivars (S49XT39, AG53X0, and S52XT08) that showed contrasting response to inoculation with Bv AP193 and OP and a non-nodulating soybean cultivar (Lee) as a check to measure nitrogen fixation. The soil had the same properties as shown above and was fertilized consequently. Artificial LED light (800 μmol mol^−1^ PAR) was used to maintain a photoperiod of 14 h of light and 8 h of darkness. Temperatures in the greenhouse ranged from 18 to 25 °C during the day and 10 to 20 °C at night. Each pot received 1000 mL of water every 2 days. Each week plants were sprayed with Tundra EC (Winfield/AgriSolutions), Talstar-Pro (FMC), and Kontos (OHP, Inc.) to prevent and control insect infestation.

Four treatments per cultivar with five replications were prepared and applied to the seeds: (1) Non-inoculated control (NI) prepared by adding water, (2) Bv AP193 alone, (3) OP alone and (4) Bv AP193 plus OP. Bv AP193 spores were prepared at a final concentration of 1 × 10^6^ CFU/50 µL per seed. Orange peel powder solution was prepared at a final concentration of 10 mg/200 µL per seed. At sowing, five seeds were evenly placed 2.5 cm below the soil surface. Each seed received the follow inoculations according to the treatment group: (1) 250 µL of sterile water, (2) 50 µL of Bv AP193 spores and 200 µL of distilled water, (3) 200 µL of orange peel powder solution and 50 µL of water and (4) first 200 µL of orange peel powder solution and then 50 µL of Bv AP193 spores. The planting method was performed as in the preliminary experiment explained above. 

#### 2.2.2. Physiological and Growth Measurements

When the plants reached the R5 developmental growth stage (~60 days after planting), SPAD values, a proxy for chlorophyll concentration, were measured using a SPAD-502 (Konica Minolta Inc., Tokyo, Japan). Midday leaf photosynthesis and stomatal conductance was also measured at R5 developmental growth stage (beginning of pod filling, ~60 days after planting, Fehr et al., [36]) on the youngest fully expanded trifoliate leaf in the top of the main stem during 10:30 am to 2:00 pm using two or three sets of LI-6400XT Portable Photosynthesis System (LI-COR Biosciences, Lincoln, NE, USA). Leaf chamber environmental conditions were adapted to meet outside environmental conditions of that day such as light intensity (1500 µmol mol^−1^ PAR), temperature (28 °C) and relative humidity (65%). 

Maximum rates of Rubisco carboxylation (Vc_max_) and Ribulose 1,5-bisphosphate (RuBP) regeneration rate (J_max_) were estimated from the response of photosynthesis to intercellular (CO_2_) (C_i_) as previously described [37]. Briefly, A-c_i_ curves were measured when plants were at the beginning of seed filling (R5) according to growth stages defined by Fehr et al. [36]. Photosynthesis was initially measured at the growth (CO_2_) (ambient, 410 ppm), and then (CO_2_) was reduced stepwise to the lowest concentration of 50 ppm, followed by a stepwise increase to the highest concentration of 1500 ppm. A total of 11 measurements per curve were recorded. During measurements, the block temperature was set at 28 °C and PPFD was set at saturated light conditions (1750 μmol m^−2^ s^−1^). Variables Vc_max_ and J_max_ were calculated using equations developed by Sharkey et al. [38].

To measure total canopy photosynthesis, a modular transparent custom chamber was designed as a closed system according to Soba et al. [39]. In summary, the chamber consisted of a base module to hold the container and seal the chamber, an intermediate transparent module to adjust chamber height, and a top module with ceiling and all sensors and tube fittings. Both the middle and top modules had four fans to ensure air mixing. The top module contains a temperature sensor (LI-1000-8, LI-COR Biosciences, Lincoln, NE, USA) placed under the side frame, a PAR sensor (LI-190, LI-COR Bioscience, Lincoln, NE, USA) on top of the frame and 5 m of polytetra-fluoroethylene (PTFE) tubing that connects the custom chamber inlet and outlet fittings to the LI-8100 (LI-COR Bioscience, Lincoln, NE, USA) that serves as a CO_2_ analyzer. For purposes of this study, CO_2_ fluxes were calculated as temporal changes in CO_2_ concentration of air passing through a closed loop in the canopy chamber. Measurements were performed within 90 s to avoid chamber over-heating. Temperatures were not observed to increase more than 1 °C during measurements. The CO_2_ evolution data were analyzed using Soil-Flux-Pro software (LI-COR Biosciences, Lincoln, NE, USA) by fitting a linear regression line to the CO_2_ evolution in the chamber, which provides a normalized sum of square residuals of the fits and R^2^ values. 

After the physiological measurements were done, aboveground plant organs were separated and total aboveground biomass (g plant^−1^) was calculated by separating leaves, stems and pods and drying them at 60 °C for at least 72 h and afterwards weighting them in a precision scale. Before drying, total leaf area (cm^2^ plant^−1^) was calculated by passing each trifoliated leaf through an LI-3000 Leaf Area Meter (LI-COR Biosciences, Lincoln, NE, USA). 

The total aboveground biomass, including leaves stems and pods, was ground to pass a 1 mm screen, weighed into tin capsules and shipped to the UC-Davis Stable Isotopes Facility (Davis, CA, USA) for ^15^N isotope analysis. Samples were analyzed using an isotope ratio mass spectrometer (IsoPrime, Elementar France, Villeurbanne) coupled to an elemental analyzer (EA3000, EuroVector, Milan, Italy). The natural ^15^N isotopic ratio (δ^15^N) in the aboveground biomass was calculated using the formula described by Shearer and Kohl [40]:$$\delta^{15}N=\frac{R_{sample}}{\left(R_{air}-1\right)}\times 1000$$
where *R_sample_* and *R_air_* are the isotope ratios (^15^*N*/^14^*N*) of the sample and air, respectively. The proportion of N derived from the atmosphere (%Ndfa), estimating the biological nitrogen fixation, was determined by the ^15^N natural abundance method [40], following the formula: $$\mathrm{Ndfa}\;\left(\%\right)=\frac{\delta^{15}N_{\mathrm{ref}}-{\;\delta }^{15}N_{\mathrm{soy}}}{\delta^{15}N_{\mathrm{soy}}-B}\times 100,$$
where Ndfa (%) is the percentage of N_2_ coming from the atmosphere through BNF; δ^15^N_ref_ is the δ^15^N signature of the non-fixing soybean reference (cultivar Lee) aboveground biomass, δ^15^N_soy_: δ^15^N is the signature of the aboveground biomass for each treatment; and B is the δ^15^N value of a soybean plant growing in a N free media relying only on BNF as source of N. The B-value used in our study was obtained as the δ^15^N average value (−1.86‰) from previous reports for soybean growing in greenhouse conditions (Appendix A).

Roots were cleaned after harvesting using tap water and the nodules were separated. Nodule and root characteristics were measured as described in the section above. 

#### 2.2.3. Statistical Analysis

Two-way ANOVA was performed for each parameter to test the effect of inoculation (Control, Bv AP193 alone, OP alone, AP193+OP), cultivar (S49XT39, AG53X0 and S52XT08) and their interaction. Two-way ANOVA, with inoculation and cultivars as main factors and replication as random variable, was performed using PROC GLIMMIX in SAS (SAS 9.4, SAS Institute, Cary, NC, USA). When the main effect of inoculation and/or genotype, or their interaction was significant, least square means post hoc tests were performed to compare means (LSMEANS, SAS 9.4, SAS Institute, Cary, NC, USA). 

### 2.3. Field Experiment to Test the Response of Inoculation with PGPR plus Orange Peel Amendment on Contrasting Soybean Cultivars

#### 2.3.1. Field Experimental Design and Inoculation Treatments 

During the Summer 2020, field trials were established at two different locations: E.V. Smith Research Center (EVS; Shorter, AL, USA) and Tennessee Valley Research Center (TV; Madison, AL, USA) in a no-tillage system, with rye as winter cover crop. E.V. Smith Research Center has a Compass loamy sand with a pH of 6.2 and soil composition of 18 kg ha^−1^ P, 88 kg ha^−1^ K, 150 kg ha^−1^ Mg, 923 kg ha^−1^ Ca. The fertilizer recommendations for soybean were 117 kg ha^−1^ of P_2_O_5_ and 184 kg ha^−1^ of K_2_O. At E.V. Smith Research Center, the mean, maximum and minimum temperature during the growing season was 22.9, 33.4 and 11 °C, respectively, with a rainfall accumulation of 887.73 mm during the growing season. Tennessee Valley Research Center has a Decatur Silt Loam soil texture with a pH of 6.5 and soil composition of 62 kg ha^−1^ P, 313 kg ha^−1^ K, 163 kg ha^−1^ Mg, 2950 kg ha^−1^ Ca. The field was not fertilized as the recommendations for soybean from the soil testing laboratory did not recommend any fertilizer application. In the Tennessee Valley Research Center, the mean, maximum and minimum temperature during the growing season was 20.9, 32.1, and 8.8 °C, respectively, with a rainfall accumulation of 809.24 mm during the growing season. Pre-emergence and post emergence herbicides and pesticides were applied following the recommendations of the Alabama Cooperative Extension System for each field. 

A randomized complete block design was used for these experiments. In total, four commercial soybean cultivars (S49XT39, AG53X0, S52XT08 and AG69X0) and a non-nodulating soybean cultivar (Williams 82 NN), as a check for the nitrogen fixation, were evaluated. Four different inoculations were applied at sowing time: 1) Non-inoculated (NI), (2) Bv AP193, (3) OP or (4) Bv plus OP. At sowing, a Bv spore suspension at 1 × 10^6^ spore CFU/mL and orange peel liquid suspension (1%) was applied in furrow in the two middle rows to avoid cross plot contamination at the rate of 37.85 L per hectare, according to sprayer specifications and following the protocol of Hassan et al. [27]. The experimental design had four replicates, with a total of 80 plots at each location. Plots were 20 foot long and consisted of four rows with 36 inches spacing between rows. In both locations, the planting density was 214,800 seed ha^−1^ with a germination percentage higher than 90%. Seeds were planted with a four row Almaco Cone Planter (Almaco Inc., Nevada, IA, USA).

#### 2.3.2. Physiological Measurements

When the plants had reached the R2 developmental growth stage (Flowering [10]), SPAD values, a proxy for chlorophyll concentration, were measured using a SPAD-502. Midday leaf photosynthesis and stomatal conductance was measured at R3 developmental growth stage (First pod [36]) in two plants per plot on the youngest fully expanded trifoliate leaf in the top of the main stem during 10:30 am to 2:00 pm using two sets of LI-6400XT Portable Photosynthesis System (LI-COR Biosciences, Lincoln NE, USA). Leaf chamber environmental conditions were adapted to meet outside environmental conditions of that day and location such as light intensity, temperature and relative humidity.

#### 2.3.3. Growth Parameters and %Ndfa Calculation

Emergence fifteen days after planting was counted twice per plot as number of seedlings per meter to estimate the percentage of germination. Plant height (cm) at R2, R5 and R7 was measured in three plants per plot from the soil surface to the apical meristem of the main stem. At pod filling (R3 [36]) aboveground biomass accumulation was measured by harvesting a total of 0.5 m where the stems emerge from the soil and dried for 72 h in an industrial forced air heating oven at 60 °C and later weighted on a precision scale. 

The total aboveground biomass, including leaves, stems and pods, was ground to pass a 1 mm screen, weighed into tin capsules and shipped to the UC-Davis Stable Isotopes Facility (Davis, California, USA) for ^15^N isotope analysis. The nitrogen derived from the atmosphere (Ndfa %) was calculated as described above using the cultivar Williams 82 NN as non-nodulating control and a B value of δ^15^N = −2.78‰ from previous reports for soybean sampled around R1-R2 developmental stage (Appendix A).

Root surface area (cm^2^) and root volume (cm^3^) at the beginning of the pod developmental stage (R3 [36]) were measured by collecting two roots per plot using the shovelomic method [41] and stored into a plastic bag in a container with ice. The roots were photographed and then analyzed for root parameters using RhizoVisionExplorer (version 2.0.3) software and set up [41].

#### 2.3.4. Statistical Analysis

Two-way ANOVA was performed for each parameter to test the effect of inoculation (Control, Bv AP193 alone, OP alone, AP193+OP), cultivar (S49XT39, AG53X0 and S52XT08) and their interaction independently in each location. Two-way ANOVA, with inoculation and cultivars as main factors and replication as a random variable, was performed using PROC GLIMMIX in SAS (SAS 9.4, SAS Institute, Cary, NC, USA). When the main effect of inoculation treatment and/or genotype, or their interaction was significant, least square means post hoc tests were performed to compare means (LSMEANS, SAS 9.4, SAS Institute, Cary, NC, USA).

## 3. Results

### 3.1. Preliminary Experiment to Test Soybean Cultivar Variation to OP Amendments

Among the 20 soybeans cultivars tested, the inoculation with Bv plus OP significantly increased plant height (14.3%), leaf area (11.4%) and total aboveground dry weight (13.2%) compared with the non-inoculated treatment (Table 1). Additionally, there was a significant effect of the cultivar variable for those parameters but there was no significant cultivar–inoculation interaction (Table 1). Despite of the lack of cultivar–inoculation interaction, the Bv+OP inoculation had a negative impact on plant growth parameters for the cultivar S54XT17, reducing plant height (3%), leaf area (16.5%) and aboveground biomass (15.2%) in contrast with the control treatment. For cultivars AG53X0, LS5588X and REV4940X the inoculation with Bv+OP also reduced the leaf area and aboveground dry weight (Table 1). 

On the other hand, the cultivars G4190RX and S49XT39 had the highest increase in plant height due to the inoculation (50.9% and 38.7%, respectively). For leaf area, cultivars AG69X0, G4190RX, S49XT39 and S52XT08 showed more than 25% increase in inoculated treatment. The cultivar S49XT39 can be highlighted with an 87.3% increase on leaf area with Bv+OP treatment compared with the non-inoculated control. Cultivars S49XT39 and S52XT08 showed a 69.8% and 31.6% increase in dry weight, respectively, with the inoculation treatment (Table 1).

The inoculation with Bv+OP significantly increased nodule number (22.9%), nodule area (26.4%), nodule dry weight (40.5%), root length (16.5%) and root dry weight (12.5%) (Table 2 and Table 3). For all the nodulation and root growth parameters measured, there was a significant cultivar effect. Additionally, only for the nodule area parameter, there was a significant effect with the cultivar–inoculation interaction (Table 2).

For the effect of inoculation on cultivar, as in the aboveground parameters, the cultivar S49XT39 was notable for the observed 163.1% increase in nodule numbers, 166.3% in nodule area, 275.4% in nodule dry weight, 45.0% in root length and 73.2% in root dry weight when compared with the NI treatment. The cultivar AG69X0 also showed a 67.5, 101.8, 176.5, 18.7 and 31.3% increase in nodule number, nodule area, nodule dry weight, root area and root dry weight, respectively, with the inoculation. On the other hand, as shown for the aboveground parameters, the inoculation with Bv+OP had a negative impact on the cultivar S54XT17, reducing the nodule number (30.8%), nodule area (34.4%), nodule dry weight (22.9%) and root area (3.9%) relative to non-inoculated plants. Moreover, there was a decrease in the root growth for AG53X0 and REV4940X when inoculated with Bv plus OP (Table 3). These contrasting results showed that although there was no significant interaction between cultivar and inoculation treatment, the response to inoculation seems to be cultivar dependent as it was observed that some cultivars responded positively while others had a negative response to the synbiotic treatment. 

Based on these data, we selected three cultivars considered responsive (S49XT39, S52XT08, AG69X0) and one non-responsive (AG53X0) to the synbiotic treatment to study the physiological response of soybean genotype to Bv plus OP inoculation to better understand cultivar variations to inoculation and the factors that can influence this response.

### 3.2. Greenhouse Experiment to Test the Response of Inoculation with PGPR plus Orange Peel Amendment on Contrasting Soybean Cultivars

#### 3.2.1. Growth Parameters

No significant inoculation effect was observed for any of the aboveground plant parameters analyzed in this experiment (Table 4). However, the effect of the cultivar–inoculation interaction was significant for leaf area and aboveground dry weight parameters (Table 4). For the cultivar itself, significant effects were observed for plant height, leaf area and pod dry weight (Table 4). 

There was an increase of leaf area with the inoculation AP193+OP compared with the non-inoculated control (NI) for cultivars AG53X0 (+20.7%) and S49XT39 (+17.2%). In contrast, the inoculation with AP193 plus OP had a negative impact for S52XT08 reducing leaf area by 44.5%, which resulted in a 10.2% decrease of the total aboveground dry weight compared to the non-inoculated control treatment (Table 4).

The S49XT39 cultivar inoculated with AP193 resulted in higher pod (+42.8%) and aboveground biomass (+17.1%; Table 4). The supplement of OP to the inoculation with AP193 did not improve the pod and aboveground dry weights for this cultivar. For cultivar AG53X0, there was no significant effect of the Bv AP193 and OP inoculation on pod dry weight; however, the AP193+OP treatment significantly increased the total aboveground biomass in 5.50 g (+31.8%) in comparison with the non-inoculated treatment (Table 4). 

There was a significant effect of the cultivar on nodule number, area and dry weight (Table 5). However, there was no effect of inoculation or the interaction of cultivar–inoculation on the nodulation and nitrogen fixation. In general, AP193 supplemented with OP reduced the nodule number, nodule area and dry weight when compared with the control (non-inoculated) treatment (Table 5). In contrast, this treatment increased nodule size (+5.2%) and the nitrogen derived from the air (+8.7%) compared with the control (Table 6). In cultivar AG53X0, the inoculation with AP193+OP showed no positive response on nodulation and nitrogen fixation parameters. The S52XT08 cultivar showed a negative response on nodulation but no significant increase in nitrogen derived from the air (4.7%) compared with the non-inoculated treatment. In contrast, there was a strong positive response of the AP193+OP inoculation on cultivar S49XT39 with an increase of 71.2% in nodule number, 65.4% in nodule area, 60.7% in nodule dry weight and 32.3% in nitrogen derived from the air in comparison with the control treatment (Table 6). 

#### 3.2.2. Photosynthesis Parameters

There was a significant cultivar effect on stomatal conductance (g_s_), canopy photosynthesis and intrinsic water-use efficiency (WUE_i_; Figure 1). The effect of the inoculation and the interaction between factors was significant for both g_s_ and WUE_i_, while it was not significant for photosynthetic rate (A). The g_s_ was significantly higher (87.3% increase) for the non-inoculated treatment in comparison with plants inoculated with AP193+OP. This resulted in superior WUE_i_ (A/g_s_) for the inoculated plants with AP193+OP in comparison with the control (38.2%). In cultivar S52XT08 the treatment with AP193+OP increased the WUE_i_ by 113.5% in comparison to the NI treatment. 

For canopy photosynthesis, the control treatment had higher flux (29.1%) compared with the AP193 plus OP inoculation.

### 3.3. Field Experiment to Test the Response of Inoculation with PGPR plus Orange Peel Amendment on Contrasting Soybean Cultivars

A cultivar effect was observed for plant height at both E.V. Smith (EVS) and Tennessee Valley (TV) locations (Figure 2). However, there was no effect of inoculation or the interaction between variables for the plant growth parameters measured. The inoculation with AP193 resulted in higher biomass accumulation (6.1%) at EVS, while at TV the OP was responsible for the highest value (5.1%) compared with the non-inoculated treatment. For plant height, plants maintained the same range on the treatments within cultivars. Cultivar AG53X0 had the total aboveground biomass (+4.2% at EVS and +20.9% at TV) and plant height (+1.6% at EVS and +3.8% at TV) increased with inoculation AP193+OP compared with the non-inoculated treatment at both locations. Therefore, this cultivar was observed to have more consistent positive results in comparison with the other cultivars as some increased growth in one location and decreased it in another. 

There was a slight inoculation effect only for nitrogen derived from the air (Ndfa) at EVS (Figure 3). However, no significant response of the interaction between variables on yield and nitrogen fixation was observed at either location. Individually, cultivar S52XT08 showed the highest Ndfa on the inoculation with AP193+OP (49.73% at EVS and 62.63% at TV), which represents a 24% (EVS) and 14% (TV) increase in comparison with the non-inoculated treatment, although this difference was not significant (Figure 3). The other cultivars showed a reduction on the nitrogen fixation with the inoculation (AP193+OP) in both locations in relation to the control treatment (non-inoculated). 

The cultivar AG69X0 showed the greatest yield for the inoculated treatment with AP193+OP (3.2 ton/ha at EVS and 4.1 ton/haat TV), which was a 14.9% (EVS) and 4.1% (TV) yield increase in comparison with the non-inoculated treatment (Figure 3). For both locations, the inoculations resulted in reduction of yield for S49XT39. For AG53X0 and S52XT08, there was a decrease in yield at EVS and an increase at TV for the inoculated treatments (Figure 3). Therefore, for yield gain, AG69X0 was the cultivar with more consistent positive responses to the inoculation with AP193+OP at different environmental conditions. 

## 4. Discussion

The effect of soybean seed inoculation with Bv AP193 plus OP was previously tested with positive results in greenhouse and field experiments but only for a one-year experiment and with one soybean cultivar [27,42]. For that reason, this study explored the response of 20 soybean cultivars to inoculation with Bv AP193 supplemented with OP as a seed treatment to assess the consistency of the synbiotic inoculant in promoting plant growth. Our results demonstrate that there was a statistically significant positive effect of the inoculation with PGPR plus OP on plant growth promotion of 13.2% for aboveground biomass (Table 2), 40% for nodule dry weight (Table 3) and 12.5% for root dry weight (Table 4) when the response was averaged across all cultivars. In addition, this significant inoculation effect occurred in the absence of inoculation by cultivar interaction. If these positive results are translated to an improvement in seedling vigor and later yield in the field, this PGPR plus OP inoculation treatment could have a very significant impact on soybean crop production and contribute to yield improvement.

However, when the effect of inoculation is analyzed by percentage of change for each cultivar, we observed that the response to the synbiotic inoculation was highly cultivar specific. Cultivars S49XT39, S52XT08, G4190RX and AG69XT0 were found to have a positive response to the inoculation with several plant growth parameters observed to increase; in contrast, cultivars AG53X0, REV4940X, LS5588X and S54XT17 showed a negative response to the inoculation (Table 2, Table 3 and Table 4). For example, the inoculation with AP193 plus OP increased root dry weight by 73 and 28% in S49XT39 and S52XT08 cultivars, respectively, while decreasing 12% in cultivar AG53X0 (Table 4). This phenomenon could be caused by different compatibility between cultivars and the PGPR strain that is derived from the capability of the PGPR strain to metabolize and use specific root exudates that can vary between the cultivar within each crop species [43], as well as the presence of plant pathogens that could metabolize pectin and the relative susceptibility of the soybean cultivars to those pathogens. It has been found that rhizosphere populations change depending on the soybean cultivar planted [44]; therefore, it is possible that PGPR compatibility and effectiveness in promoting growth can change with the cultivar of soybean tested. Similarly to our study, Kuzmicheva et al. [45] found that inoculation with *Pseudomonas oryzihabitans* (strain Ep4) stimulated root growth of the soybean cultivars Nice-Mecha and Svapa, which produced more organic acids; meanwhile, the cultivar Bara that secreted less organic acids did not show root growth promotion. While our preliminary results showed a positive trend of inoculation when all the cultivars response was averaged, we have found that some cultivars did not respond to inoculations as demonstrated in other studies [43,45,46,47,48]. These cultivar by PGPR differential responses could be a constraint for inoculant manufacturers since there is a need for consistently effective biologic inoculants that can broadly be used in agriculture. For that reason, the factors that can affect the genotype by PGPR strain responses need to be further studied. 

The different response of the cultivars to the inoculation could also be explained by the response of each cultivar to the OP amendment used in this preliminary study. While carrier materials can increase plant growth in combination with PGPR, the inoculation method can also cause stress to the introduced bacteria [49]. Furthermore, OP has a complex chemical composition, including phenolic fractions that could affect the PGPR and/or each cultivar performance [27,42]. For that reason, the greenhouse experiment investigated the combined and separate effects of PGPR and OP on a set of cultivars selected for their high responsiveness (S49XT39 and S52XT08) or lack of response (AG53X0) to inoculation. 

As in the preliminary greenhouse experiment, the cultivar S49XT39 showed positive plant growth promotion after inoculation with AP193 plus OP with a 9%, 60% and 32% increase on aboveground biomass, nodule dry weight and nitrogen fixation, respectively, in comparison with the non-inoculated treatment (Table 5 and Table 6). In contrast, the cultivar AG53X0 that did not show an increase in aboveground biomass with the inoculation in the preliminary experiment showed a 31.8% increase in biomass in this experiment. Additionally, one of the other responsive cultivars during the preliminary experiment, S52XT08, showed a negative response to the inoculation. This lack of consistency between experiments could be due to a strong influence of the environment and/or soil microbiota on the plant response to PGPR and OP inoculation. According to Nadeem et al. [50], the effectiveness of inoculation with PGPRs on plant growth promotion might vary depending on microbial populations and their interactions with environmental factors such as soil nutrition, moisture and temperature. Since our preliminary experiment was conducted during the winter of 2020 and the second experiment during spring 2020 in a greenhouse, the environmental conditions such as light intensity, quality and temperature may have affected plant responses to synbiotic inoculation. Light intensity and quality can affect photosynthesis [51], which ultimately affects the amount of root exudates produced by the plant and therefore might interfere in the plant–PGPR crosstalk [43].

The greenhouse experiment also showed that inoculated plants with PGPR plus OP had lower stomatal conductance (g_s_) without any negative effect on plant photosynthesis, which leads to superior intrinsic water-use efficiency (WUE_i_). WUE_i_ is an instantaneous measurement of the efficiency of carbon gain per water loss. The WUE_i_ tended to be higher in all the cultivars inoculated with PGPR plus OP but was significantly increased in S52XT08 (Figure 1). Changes in WUE_i_ are the result of decreases in transpiration rate or increases in photosynthesis activity [52]. In maize, soil inoculation with *Burkholderia* sp. LD-_11_ also improved WUE_i_ through reduction of stomatal aperture provoked by small increases in abscisic acid (ABA) concentration in the leaves, which also promoted biomass accumulation [53]. While in our current experiment we did not measure ABA concentrations, we hypothesize that inoculation with Bv AP193 may produce an increase in WUE_i_ due to ABA production. These results are supported by peanut experiments performed also at Auburn University where the inoculation with Bv AP203 with OP amendment resulted in increased WUE_i_ under well-watered and drought stress conditions [42]. These results indicate that inoculation with some PGPR strains plus OP may be an important tool to alleviate water stress and benefit plant survival under water shortage environments. 

In the field trials, there was no significant effect of inoculation and the interaction between variables for plant growth, nitrogen fixation and yield (Figure 2 and Figure 3). While non-significant, cultivar AG69X0 presented the greatest yield on the PGPR plus OP treatment with 14.9% (EVS) and 4.1% (TV) yield increases in comparison with the non-inoculated control. While not significant, these increases are considered “acceptable” by the farmers and by inoculant manufacturers [12] and therefore should be further studied for commercial application. On the other hand, the other two cultivars showed no effects or even decreases in yield at both locations (Figure 3). As significant positive effects have been observed in some of the greenhouse experiments, the lack of a significant effect in the field experiment could be attributed to: (1) Effects of the environment [43,51]; (2) competition with soil indigenous microbiota [22,54,55,56]; (3) influence of parasites and pathogens [22,57]; or (4) leaching of the inoculum and amendment due to the spray application and the occurrence of extreme precipitation events. While Bv AP193 was selected for its capability to grow and consume pectin [27], if the pectin washed away or was diluted in the soil, this may explain the lack of significant effects observed in the field. It could also be that there may be other *Bacillus* strains or pathogenic microorganisms that were able to grow more rapidly on a pectin-rich substrate and therefore may be able to survive better under field conditions. Future experiments should focus on isolating new *Bacillus* strains that catabolize pectin-rich substrates rapidly, or adapting existing strains to improve their ability to grow on OP as a growth substrate, in order to achieve a better plant response. In addition, it needs to be tested whether seed coating or in-furrow seed treatment will be more effective in producing a stable growth promotion response in multiple cultivars under different environments. 

In this experiment we have demonstrated that inoculation with AP193 plus orange peel increases soybean growth characteristics of a broad variety of cultivars. However, when the effect of inoculation is analyzed individually cultivar by cultivar, some cultivars showed a very positive significant effect of the inoculation, while other did not showed any effect. In order for this treatment to be usable by industry, the response of all cultivars needs to be homogeneous and therefore the variable response obtained in our experiment needs to be further investigated. Chemical analysis of root exudates of responsive and irresponsive cultivars could inform us of the root exudates that are more effective at attracting and promoting the growth of specific PGPRs. Likewise, transcriptome studies of specific cultivars and PGPR strains could give us information about the molecular mechanisms that mediate plant–microbe interactions in the context of a pectin-rich amendment. Lastly, research in new inoculation methods and synbiotic formulations could facilitate better outcomes for PGPR–plant responses across a wider diversity of cultivars and environmental conditions. 

## 5. Conclusions

A greenhouse experiment showed a significant positive effect of the inoculation with AP193 plus orange peel on soybean growth promotion, when analyzed for cultivar-specific responses. However, we observed cultivars with a very positive response and some with no response to inoculation. The cultivar-specific responses may be explained by cultivar–strain crosstalk, where the PGPR is able or not to degrade and use the root exudates as C and energy source, inoculation method used and/or orange peel composition. Furthermore, we noticed a lack of consistency of results when analyzing the cultivars selected from the preliminary experiment. This can be due to a strong influence of the environment on the plant response to the PGPR inoculation, such as soil nutrient status, moisture, temperature and/or light intensity. Moreover, the competition of the introduced PGPR strain with the soil native bacteria can influence PGPR survival in the rhizosphere, thereby reducing their beneficial effect on plant growth promotion. Further studies are needed to assess the factors that can affect the communication between soybean cultivar and PGPR and to identify ways to enhance the efficacy of a synbiotic treatment in promoting plant growth. 

## Figures and Tables

**Figure 1 plants-11-01138-f001:**
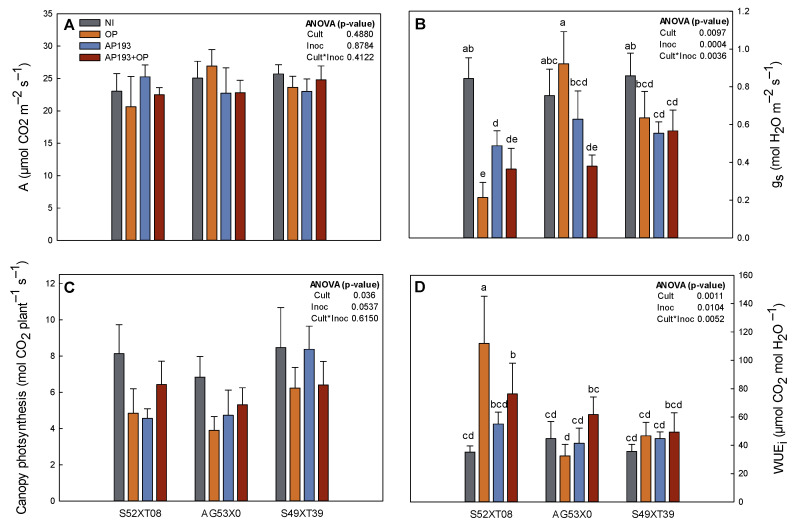
Photosynthesis rate (A, **A**), stomatal conductance (gs, **B**), canopy photosynthesis (**C**) and intrinsiceffective water-use efficiency (WUEi, **D**) measured at R4 for three soybeans cultivars (AG53X0, S49XT39, S52XT08) grown in the greenhouse with four different inoculations: Non-inoculated control (NI), orange peel alone (OP), Bacillus velezensis strain AP193 alone (AP193), and the combination of Bacillus velezensis strain AP193 and orange peel (AP193+OP). Bars represent the standard error for each treatment. Different letters between gs and WUEi represent treatments that were statistically different (*p*-value < 0.05).

**Figure 2 plants-11-01138-f002:**
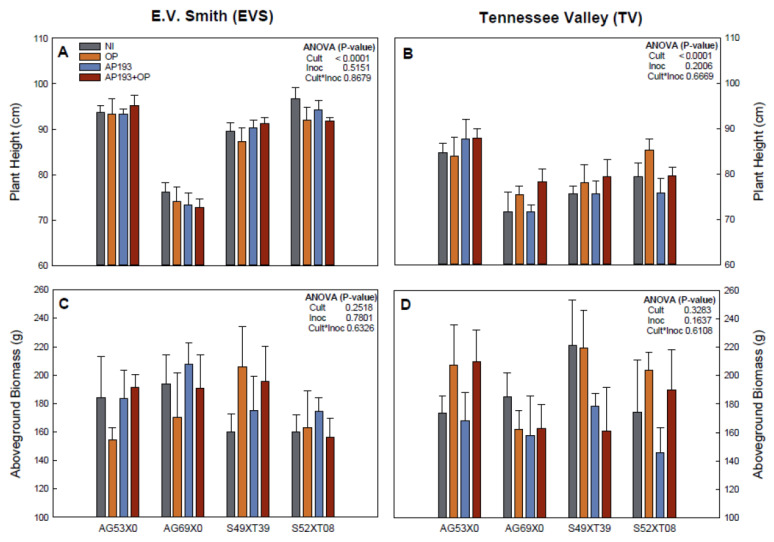
Plant height (**A**,**B**) and total aboveground dry biomass (**C**,**D**) for four different soybean cultivars grown at field conditions during Summer 2020 at two locations in Alabama, USA (E.V. Smith and Tennessee Valley). Four inoculation treatments were tested at sowing time: Non-inoculated control (NI), orange peel alone (OP), Bacillus velezensis strain AP193 alone (AP193),and the combination of Bacillus velezensis strain AP193 and orange peel (AP193+OP). Bars represent the standard error for each treatment.

**Figure 3 plants-11-01138-f003:**
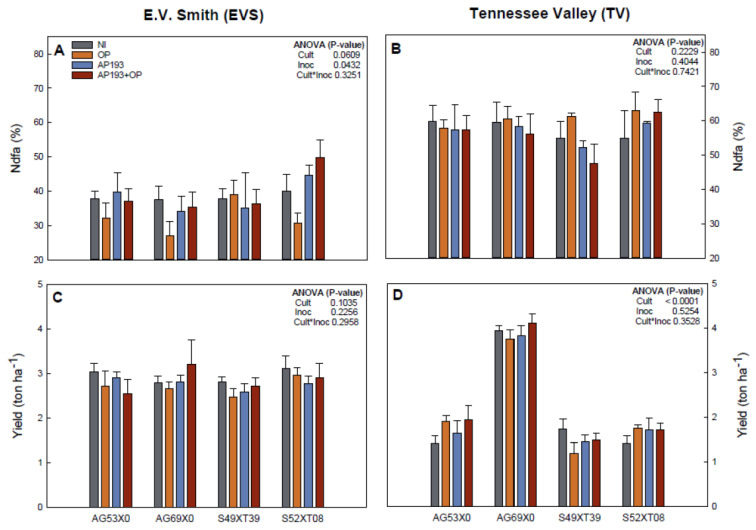
Percentage of nitrogen derived from the air (Ndfa, %, **A**,**B**) and yield (**C**,**D**) for four different soybean cultivars grown at field conditions during Summer 2020 at two locations in Alabama, USA (E.V. Smith and Tennessee Valley). Four inoculation treatments were tested at sowing time: Non-inoculated control (NI), orange peel alone (OP), Bacillus velezensis strain AP193 alone (AP193) and the combination of Bacillus velezensis strain AP193 and orange peel (AP193+OP). Bars represent the standard error for each treatment.

**Table 1 plants-11-01138-t001:** Response of inoculation with Bv AP193 plus OP on plant height, leaf area and total aboveground biomass in a preliminary greenhouse experiment with 20 soybean cultivars. The bottom section of the table shows the ANOVA results (*p*-value) for the effect of cultivar, inoculation and the interaction cultivar–inoculation. Different letters indicate significant differences within the treatment group.

Cultivar	Inoculation	Plant Height (cm)		Leaf Area (cm^2^)		Total Aboveground Dry Weight (g)	
		Estimate	% Change	Estimate	% Change	Estimate	% Change
**AG44X0**	**NI**	16.3	14.0	142.6	10.9	1.5	22.7
	**Bv+OP**	18.6		158.2		1.8	
**AG53X0**	**NI**	19.8	3.5	192.6	−5.5	2.0	−12.9
	**Bv+OP**	20.5		181.9		1.8	
**AG69X0**	**NI**	37.4	10.9	172.3	33.1	1.7	22.3
	**Bv+OP**	41.5		229.3		2.1	
**CZ 4539GTLL**	**NI**	14.1	14.0	116.6	7.0	1.2	12.7
	**Bv+OP**	16.1		124.7		1.4	
**CZ 5859LL**	**NI**	30.1	7.4	236.5	9.5	1.9	10.0
	**Bv+OP**	32.3		258.9		2.1	
**CZ 6515LL**	**NI**	22.6	18.9	149.5	22.0	1.6	27.0
	**Bv+OP**	26.8		182.3		2.1	
**G4190RX**	**NI**	14.5	50.8	134.1	26.7	1.5	25.4
	**Bv+OP**	21.9		170.0		1.9	
**G5000RX**	**NI**	16.6	9.9	150.5	13.4	1.5	7.8
	**Bv+OP**	18.3		170.6		1.6	
**GoSoy 512E18**	**NI**	25.9	25.1	161.5	20.1	1.6	20.1
	**Bv+OP**	32.5		194.0		1.9	
**LS4798X**	**NI**	19.2	14.2	160.4	4.2	1.6	13.4
	**Bv+OP**	22.0		167.1		1.9	
**LS5087X**	**NI**	16.7	26.2	155.5	18.7	1.4	22.4
	**Bv+OP**	21.1		184.6		1.7	
**LS5588X**	**NI**	29.6	5.7	283.7	−7.3	1.9	−3.5
	**Bv+OP**	31.3		262.9		1.9	
**LSX6501XS**	**NI**	20.4	8.7	183.6	16.9	1.6	18.5
	**Bv+OP**	22.2		214.6		1.8	
**NKS49-F5X**	**NI**	18.0	9.9	164.0	8.8	1.7	9.6
	**Bv+OP**	19.8		178.5		1.9	
**REV 4940X**	**NI**	17.4	14.7	148.6	−8.1	1.3	−3.3
	**Bv+OP**	20.0		136.5		1.3	
**REV 5659X**	**NI**	28.1	22.8	259.6	14.2	1.9	14.5
	**Bv+OP**	34.5		296.6		2.1	
**S49XT39**	**NI**	17.1	38.7	108.2	87.3	1.0	69.8
	**Bv+OP**	23.7		202.6		1.7	
**S52XT08**	**NI**	19.5	22.8	192.9	26.6	1.7	31.6
	**Bv+OP**	23.9		244.3		2.2	
**S54XT17**	**NI**	33.9	−3.0	280.6	−16.5	2.0	−15.2
	**Bv+OP**	32.9		234.2		1.7	
**S56XT99**	**NI**	32.9	6.0	240.4	6.1	1.8	8.1
	**Bv+OP**	34.9		255.0		2.0	
**NI Mean**		22.5 b	14.3	181.7 b	11.4	1.6 b	13.2
**Bv+OP Mean**		25.7 a		202.4 a		1.8 a	
**2-WAY ANOVA RESULTS**							
Cultivar (*p*-value)		<0.0001		<0.0001		0.0361	
Inoculation (*p*-value)		0.0002		0.0107		0.0032	
Cultivar–inoculation		0.994		0.7378		0.8728	

NI—non-inoculated; Bv+OP—*Bacillus velezensis* plus orange peel.

**Table 2 plants-11-01138-t002:** Response of inoculation with Bv AP193 plus OP on nodule number, nodule area and nodule dry weight in a preliminary greenhouse experiment with 20 soybean cultivars. The bottom section of the table shows the ANOVA results (*p*-value) for the effect of cultivar, inoculation and the interaction cultivar–inoculation. Different letters indicate significant differences within the treatment group.

Cultivar	Inoculation	Nodule Number		Nodule Area (cm^2^)		Nodule Dry Weight (g)	
		Estimate	% Change	Estimate	% Change	Estimate	% Change
**AG44X0**	**NI**	36.2	−3.4	2.2	1.9	0.045	18.3
	**Bv+OP**	35.0		2.2		0.053	
**AG53X0**	**NI**	24.2	12.4	1.5	15.9	0.043	25.6
	**Bv+OP**	27.2		1.8		0.053	
**AG69X0**	**NI**	30.0	67.5	1.7	101.8	0.041	176.5
	**Bv+OP**	50.2		3.5		0.114	
**CZ 4539GTLL**	**NI**	18.7	49.3	1.2	43.2	0.025	82.1
	**Bv+OP**	28.0		1.7		0.045	
**CZ 5859LL**	**NI**	31.2	−2.4	2.1	15.3	0.057	30.7
	**Bv+OP**	30.5		2.4		0.074	
**CZ 6515LL**	**NI**	23.9	49.8	1.8	42.1	0.051	40.3
	**Bv+OP**	35.8		2.6		0.071	
**G4190RX**	**NI**	17.6	68.7	1.3	63.1	0.032	89.8
	**Bv+OP**	29.7		2.2		0.062	
**G5000RX**	**NI**	25.7	36.9	1.8	9.5	0.052	0.0
	**Bv+OP**	35.2		2.0		0.052	
**GoSoy 512E18**	**NI**	29.7	−10.9	2.3	−5.7	0.064	7.8
	**Bv+OP**	26.5		2.1		0.069	
**LS4798X**	**NI**	32.2	11.6	1.8	17.3	0.054	−0.8
	**Bv+OP**	36.0		2.1		0.054	
**LS5087X**	**NI**	22.2	52.8	1.1	121.3	0.024	218.0
	**Bv+OP**	34.0		2.5		0.076	
**LS5588X**	**NI**	23.2	41.9	1.5	21.9	0.042	25.3
	**Bv+OP**	33.0		1.8		0.052	
**LSX6501XS**	**NI**	32.2	35.7	2.1	28.7	0.054	45.2
	**Bv+OP**	43.7		2.8		0.079	
**NKS49-F5X**	**NI**	37.0	26.3	2.3	11.1	0.061	8.8
	**Bv+OP**	46.7		2.6		0.067	
**REV 4940X**	**NI**	40.2	−32.9	2.2	−19.7	0.047	4.7
	**Bv+OP**	27.0		1.7		0.050	
**REV 5659X**	**NI**	40.2	6.2	2.3	37.8	0.049	71.2
	**Bv+OP**	42.7		3.2		0.084	
**S49XT39**	**NI**	16.2	163.1	0.9	166.3	0.016	275.4
	**Bv+OP**	42.7		2.5		0.061	
**S52XT08**	**NI**	30.0	55.0	1.6	88.5	0.040	108.5
	**Bv+OP**	46.5		3.1		0.083	
**S54XT17**	**NI**	45.5	−30.8	3.7	−34.4	0.101	−22.9
	**Bv+OP**	31.5		2.4		0.078	
**S56XT99**	**NI**	35.0	27.9	2.5	18.9	0.062	17.8
	**Bv+OP**	44.7		2.9		0.072	
**NI Mean**		29.6 b	22.9	1.9 b	26.4	0.048 b	40.5
**Bv+OP Mean**		36.3 a		2.4 a		0.067 a	
**2-WAY ANOVA RESULTS**							
Cultivar (*p*-value)		0.0007		0.0003		0.0017	
Inoculation (*p*-value)		<0.0001		<0.0001		<0.0001	
Cultivar–inoculation		0.0323		0.0171		0.0443	

NI—non-inoculated; Bv+OP—*Bacillus velezensis* plus orange peel.

**Table 3 plants-11-01138-t003:** Response of inoculation with Bv AP193 plus OP on root area, root length and root dry weight in a preliminary greenhouse experiment with 20 soybean cultivars. The bottom section of the table shows the ANOVA results (*p*-value) for the effect of cultivar, inoculation and the interaction cultivar–inoculation. Different letters indicate significant differences within the treatment group.

Cultivar	Inoculation	Root Area (cm^2^)		Root Length (cm)		Root Dry Weight (g)	
		Estimate	% Change	Estimate	% Change	Estimate	% Change
**AG44X0**	**NI**	361.6	5.8	2935.6	29.8	0.36	11.1
	**Bv+OP**	382.7		3810.7		0.40	
**AG53X0**	**NI**	417.5	−3.9	4175.2	−11.9	0.49	−12.7
	**Bv+OP**	401.1		3679.1		0.43	
**AG69X0**	**NI**	330.5	18.7	3584.6	30.4	0.40	31.3
	**Bv+OP**	392.3		4673.4		0.52	
**CZ 4539GTLL**	**NI**	326.0	9.5	2188.8	19.5	0.32	−13.1
	**Bv+OP**	356.8		2616.0		0.28	
**CZ 5859LL**	**NI**	383.3	−3.4	3896.8	3.7	0.48	1.4
	**Bv+OP**	370.3		4042.4		0.49	
**CZ 6515LL**	**NI**	374.4	6.3	3038.3	38.0	0.35	32.5
	**Bv+OP**	397.9		4192.4		0.47	
**G4190RX**	**NI**	392.3	0.0	2951.2	18.0	0.39	11.7
	**Bv+OP**	392.5		3483.8		0.44	
**G5000RX**	**NI**	351.3	7.4	2702.1	31.8	0.37	9.3
	**Bv+OP**	377.4		3562.1		0.41	
**GoSoy 512E18**	**NI**	413.6	−4.2	3078.5	19.4	0.38	16.9
	**Bv+OP**	396.1		3676.9		0.44	
**LS4798X**	**NI**	376.3	10.5	3614.3	11.0	0.37	−6.3
	**Bv+OP**	415.7		4010.9		0.34	
**LS5087X**	**NI**	416.1	−1.9	2779.3	16.9	0.30	31.2
	**Bv+OP**	408.0		3249.2		0.39	
**LS5588X**	**NI**	395.2	2.4	3958.7	11.4	0.49	2.5
	**Bv+OP**	404.6		4408.4		0.51	
**LSX6501XS**	**NI**	399.9	−2.8	3638.0	13.3	0.42	32.2
	**Bv+OP**	388.5		4122.7		0.55	
**NKS49-F5X**	**NI**	378.4	3.4	3350.6	9.7	0.42	4.8
	**Bv+OP**	391.5		3675.9		0.44	
**REV 4940X**	**NI**	362.9	−13.2	3327.2	−7.5	0.35	−4.5
	**Bv+OP**	315.0		3077.4		0.34	
**REV 5659X**	**NI**	364.1	5.2	3489.3	20.4	0.57	−4.0
	**Bv+OP**	383.1		4200.9		0.55	
**S49XT39**	**NI**	320.4	21.8	2567.2	45.0	0.24	73.2
	**Bv+OP**	390.3		3722.4		0.41	
**S52XT08**	**NI**	362.9	8.3	3535.3	24.5	0.43	28.3
	**Bv+OP**	393.2		4402.1		0.55	
**S54XT17**	**NI**	379.4	−3.8	3601.5	4.7	0.35	12.9
	**Bv+OP**	364.9		3770.3		0.39	
**S56XT99**	**NI**	375.3	12.4	4033.0	25.6	0.46	27.5
	**Bv+OP**	421.9		5064.8		0.59	
**NI Mean**		374.1	3.5	3322.3 b	16.5	0.40 b	12.5
**Bv+OP Mean**		387.2		3872.1 a		0.45 a	
**2-WAY ANOVA RESULTS**							
Cultivar (*p*-value)		0.0146		<0.0001		<0.0001	
Inoculation (*p*-value)		0.0529		<0.0001		0.0088	
Cultivar–inoculation		0.5024		0.9111		0.8248	

NI—non-inoculated; Bv+OP—*Bacillus velezensis* plus orange peel.

**Table 4 plants-11-01138-t004:** Plant height, leaf area, pod dry weight and aboveground biomass of three soybean cultivars (AG53X0, S49XT39, S52XT08) inoculated with Bv AP193, OP alone, the combination of Bv AP193 and OP or the NI control and grown under greenhouse conditions. The bottom section of the table shows the ANOVA results (*p*-value) for the effect of cultivar, inoculation, and the interaction cultivar–inoculation. Different letters indicate significant differences within the treatment group.

Cultivar	Inoculation	Plant Height (cm)	Leaf Area (cm^2^)	Pod Dry Weight (g)	Aboveground Dry Weight (g)
		Estimate	Estimate	Estimate	Estimate
**AG53X0**	**NI**	55.5	1394.2 b	7.7	17.3 bcd
	**AP193**	58.5	1451.8 b	4.7	15.0 d
	**OP**	56.4	1479.8 b	6.5	17.2 bcd
	**AP193+OP**	51.2	1682.6 b	7.6	22.8 a
**S49XT39**	**NI**	59.7	1409.1 b	5.8	16.5 bcd
	**AP193**	59.6	1498.3 b	8.3	19.3 abc
	**OP**	58.5	1270.5 b	6.3	16.0 bcd
	**AP193+OP**	58.2	1652.0 b	6.3	18.1 bcd
**S52XT08**	**NI**	54.1	2553.1 a	5.2	19.9 a
	**AP193**	45.5	1632.1 b	5.4	15.9 cd
	**OP**	48.1	2928.5 a	4.3	18.3 bcd
	**AP193+OP**	43.0	1415.8 b	5.4	17.9 bcd
**NI Mean**		56.4	1785.5	6.2	17.9
**AP193 Mean**		54.5	1527.4	6.1	16.7
**OP Mean**		54.4	1892.9	5.7	17.2
**AP193+OP Mean**		50.8	1583.5	6.4	19.6
**2-WAY ANOVA RESULTS**					
Cultivar (*p*-value)		<0.0001	0.0003	0.0354	0.8293
Inoculation (*p*-value)		0.2382	0.1839	0.8263	0.1502
Cultivar–inoculation		0.6569	0.0019	0.142	0.0343

NI—non-inoculated; AP193—*Bacillus velezensis* strain AP193; OP—orange peel.

**Table 5 plants-11-01138-t005:** Nodule number, nodule area, nodule dry weight, nodule per gram and percentage of nitrogen derived from the atmosphere (% Ndfa) of three soybean cultivars (AG53X0, S49XT39, S52XT08) inoculated with Bv AP193, OP or the combination of Bv AP193 and OP, or and the NI control and grown under greenhouse conditions. The bottom section of the table shows the ANOVA results (*p*-value) for the effect of cultivar, inoculation and the interaction cultivar–inoculation. Different letters indicate significant differences within the treatment group.

Cultivar	Inoculation	Nodule Number	Nodule Area (cm^2^)	Nodule Dry Weight (g)	Nodule Size (mm^2^)	% Ndfa
		Estimate	Estimate	Estimate	Estimate	Estimate
**AG53X0**	**NI**	219.2	12.1	0.51	5.2	48.6
	**AP193**	164.2	4.8	0.15	3.0	26.9
	**OP**	248.4	11.4	0.47	4.8	49.8
	**AP193+OP**	178.0	8.8	0.37	5.7	45.3
**S49XT39**	**NI**	166.6	7.8	0.27	4.3	41.3
	**AP193**	281.0	13.2	0.44	4.6	42.8
	**OP**	280.6	11.2	0.38	4.1	42.2
	**AP193+OP**	285.2	13.0	0.43	4.6	54.6
**S52XT08**	**NI**	367.0	17.8	0.70	4.9	54.4
	**AP193**	324.6	16.8	0.69	5.1	48.0
	**OP**	298.4	13.5	0.57	4.6	56.2
	**AP193+OP**	237.8	10.7	0.47	4.8	57.0
**NI Mean**		251.0	12.6	0.49	4.8	48.1
**AP193 Mean**		256.6	11.6	0.43	4.2	39.3
**OP Mean**		275.8	12.0	0.47	4.5	49.4
**AP193+OP Mean**		233.7	10.9	0.42	5.0	52.3
Cultivar (*p*-value)		0.0279	0.0214	0.007	0.5937	0.0677
Inoculation (*p*-value)		0.8095	0.8763	0.8456	0.4745	0.1083
Cultivar–inoculation		0.3136	0.1213	0.1633	0.2784	0.5526

NI—non-inoculated; AP193—*Bacillus velezensis* strain AP193; OP—orange peel.

**Table 6 plants-11-01138-t006:** Root area, root length and root dry weight of three soybean cultivars (AG53X0, S49XT39, S52XT08) inoculated with Bv AP193, OP the combination of Bv AP193 and OP, or the NI control and grown under greenhouse conditions. The bottom section of the table shows the ANOVA results (*p*-value) for the effect of cultivar, inoculation, and the interaction cultivar–inoculation.

Cultivar	Inoculation	Root Area (cm^2^)	Root Length (cm)	Root Dry Weight (g)
		Estimate	Estimate	Estimate
**AG53X0**	**NI**	1146.4	6609.3	2.86
	**AP193**	969.6	5543.3	2.56
	**OP**	1302.9	7673.4	3.32
	**AP193+OP**	1136.4	5635.3	3.06
**S49XT39**	**NI**	1051.1	4993.0	2.80
	**AP193**	1411.0	6979.5	3.22
	**OP**	1153.3	7559.8	2.70
	**AP193+OP**	1210.7	7305.4	2.84
**S52XT08**	**NI**	1138.9	6078.3	3.87
	**AP193**	1139.1	4635.3	3.99
	**OP**	1052.2	5308.9	3.61
	**AP193+OP**	921.7	4699.1	2.94
**NI Mean**		1112.1	5893.6	3.18
**AP193 Mean**		1173.2	5719.4	3.26
**OP Mean**		1169.5	6847.4	3.21
**AP193+OP Mean**		1089.6	5879.9	2.94
**2-WAY ANOVA RESULTS**				
Cultivar (*p*-value)		0.4616	0.0944	0.0097
Inoculation (*p*-value)		0.8963	0.5448	0.6931
Cultivar–inoculation		0.5281	0.4514	0.2593

NI—non-inoculated; AP193—*Bacillus velezensis* strain AP193; OP—orange peel.

## Data Availability

The raw data of these experiments can be provided personally by requesting them to the corresponding author (sanz@auburn.edu).

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
