# Peer review of "The Response to Inoculation with PGPR Plus Orange Peel Amendment on Soybean Is Cultivar and Environment Dependent"

_plants, 2022, doi:10.3390/plants11091138_

Round 1

Reviewer 1 Report

The authors reported and interesting and easy to read study

The introduction is well organized and written

Material and Method section

Line 422, Sandy Loam field soil… please report the chemical characteristic of the soil

Line 447, Artificial LED… please report the light intensity

Line 450, please report the fungicides used

Line 575, please report the chemical characteristic of the soil

Line 578, please report the chemical characteristic of the soil

Line 581, please report the fertilizers used and the amount

Line 583, please report the fertilizers used and the amount

Line 595, please add the seed density used, the percentage of germination of the genotype, the machine used for the sowing

Results are clear reported

Discussion is well argumented

Author Response

In the following text, we have added reviewer 1 questions in bold and our responses to reviewers in non bolded text.

Reviewer 1

The introduction is well organized and written

Material and Method section

Line 422, Sandy Loam field soil… please report the chemical characteristic of the soil

Soil nutrient concentration has not been provided as our soil laboratory only provide the pH and the amount of fertilizer recommended. 

Line 447, Artificial LED… please report the light intensity

LED light provided 800 ppm of PAR intensity. The information has been added in several parts of the material and methods. 

Line 450, please report the fungicides used

We were not able to find the reference of the fungicide in line 450 or other parts on the manuscript but we have specified the information about insecticides used in the greenhouse experiment in line 380 and line 423.

Line 575, please report the chemical characteristic of the soil

The chemical properties of the soil have been added as the reviewer recommended.

Line 578, please report the chemical characteristic of the soil

The chemical properties of the soil have been added as the reviewer recommended.

Line 581, please report the fertilizers used and the amount

The Fertilizer used in EV-Smith has been added as the reviewer suggested.

Line 583, please report the fertilizers used and the amount

We did not fertilized TV as the soil test did not ask for any fertilizer. We have pointed this in the manuscript just in case.

Line 595, please add the seed density used, the percentage of germination of the genotype, the machine used for the sowing

Information about seed population density, germination rate and planter used has been added.

Results are clear reported

Discussion is well argumented

Reviewer 2 Report

The authors proved that inoculation with Bv+OP significantly  increased several parameters which characterize plant growth but with significant cultivar effect. The results obtained are interesting but at the same time disturbing because particular cultivars can respond negatively to the synbiotic treatment. It obviously needs further investigations. I think that in 'Discussion' the authors should propose a research approaches that would indicate a possible solution to this problem.

Author Response

In the following text, we have added reviewer 2 questions in bold and our responses to reviewers in non bolded text.

Reviewer 2

The authors proved that inoculation with Bv+OP significantly  increased several parameters which characterize plant growth but with significant cultivar effect. The results obtained are interesting but at the same time disturbing because particular cultivars can respond negatively to the synbiotic treatment. It obviously needs further investigations. I think that in 'Discussion' the authors should propose a research approaches that would indicate a possible solution to this problem.

At the end of the discussion a paragraph proposing new research to increase our knowledge of cultivar variation to inoculation has been added as the reviewer 2 requested.
